# The Role of Urothelial Cancer-Associated 1 in Gynecological Cancers

Eleni Nousiopoulou [1], Kleio Vrettou [2], Christos Damaskos [1,3,*], Nikolaos Garmpis [1,3], Anna Garmpi [4], Panagiotis Tsikouras [5], Nikolaos Nikolettos [5], Konstantinos Nikolettos [5] and Iason Psilopatis [6,*]

1 Second Department of Propedeutic Surgery, Laiko General Hospital, Medical School, National and Kapodistrian University of Athens, 11527 Athens, Greece; eleni.nous27@gmail.com (E.N.); nikosg22@hotmail.com (N.G.)
2 Department of Cytopathology, Sismanogleio General Hospital, 15126 Athens, Greece; kliovr1@gmail.com
3 Nikolaos Christeas Laboratory of Experimental Surgery and Surgical Research, Medical School, National and Kapodistrian University of Athens, 11527 Athens, Greece
4 First Department of Propedeutic Internal Medicine, Laiko General Hospital, Medical School, National and Kapodistrian University of Athens, 11527 Athens, Greece; annagar@windowslive.com
5 Obstetric and Gynecologic Clinic, Medical School, Democritus University of Thrace, 68110 Alexandroupolis, Greece; ptsikour@med.duth.gr (P.T.); nnikolet@med.duth.gr (N.N.); knikolet@med.duth.gr (K.N.)
6 Universitätsklinikum Erlangen-Frauenklinik, Universitätsstraße 21/23, 91054 Erlangen, Germany
* Correspondence: x_damaskos@yahoo.gr (C.D.); iason.psilopatis@uk-erlangen.de (I.P.)

**Abstract:** Gynecological cancers (GC) represent some of the most frequently diagnosed malignancies in women worldwide. Long-non-coding RNAs (lncRNAs) are regulatory RNAs increasingly being recognized for their role in tumor progression and metastasis in various cancers. Urothelial cancer-associated 1 (UCA1) is a lncRNA, first found deregulated in bladder cancer, and many studies have exposed its oncogenic effects in more tumors since. However, the role of UCA1 in gynecological malignancies is still unclear. This review aims to analyze and define the role of UCA1 in GC, in order to identify its potential use as a diagnostic, prognostic, or therapeutic biomarker of GC. By employing the search terms "UCA1", "breast cancer", "endometrial cancer", "ovarian cancer", "cervical cancer", "vaginal cancer", and "vulvar cancer" in the PubMed database for the literature review, we identified a total of sixty-three relevant research articles published between 2014 and 2024. Although there were some opposing results, UCA1 was predominantly found to be upregulated in most of the breast, endometrial, ovarian, cervical, and vulvar cancer cells, tissue samples, and mouse xenograft models. UCA1 overexpression mainly accounts for enhanced tumor proliferation and increased drug resistance, while also being associated with some clinicopathological features, such as a high histological grade or poor prognosis. Nonetheless, no reviews were identified about the involvement of UCA1 in vaginal carcinogenesis. Therefore, further clinical trials are required to explore the role of UCA1 in these malignancies and, additionally, examine its possible application as a target for upcoming treatments, or as a novel biomarker for GC diagnosis and prognosis.

**Keywords:** urothelial cancer-associated 1; UCA1; gynecological; cancer

## 1. Introduction

Gynecological cancers (GC) can be generally divided into two categories: breast cancer (BC) and tumors of the female genital tract, such as endometrial cancer (EC), ovarian cancer (OC), cervical cancer (CC), vaginal cancer (VGC), and vulvar cancer (VC). BC is the most frequently diagnosed malignancy in women across the globe, with approximately 298,000 new cases reported in 2023 [1]. Mammography and breast examination represent the two main screening methods for BC. Histopathological examination of tumor biopsies represents the gold standard for diagnosis [2]. The latest research proves that about 70% of all cases are estrogen receptor-positive (ER+), whereas triple-negative BC (TNBC), lacking

estrogen receptors, progesterone receptors (PR), and the human epidermal growth factor receptor 2 (Her2), constitute 10–20% of BC cases [3,4]. The therapeutic approach for BC patients generally consists of chemotherapy, hormone therapy, and Her2-targeted drugs, such as trastuzumab and pertuzumab, combined with surgery and radiotherapy, depending on the staging [5].

EC is among the most prevalent tumors found in women globally, as it represents approximately 7% of all gynecological malignancies [6,7]. The median age of diagnosis is estimated at 63, considering that it mainly affects women of ages between 55 and 64, during the postmenopausal phase [7]. Based on the histopathology observed, there are two main categories of endometrial cancer: type I and type II. The first one derives from atypical endometrial hyperplasia and is directly linked to prolonged exposure to high levels of ERs. On the other hand, type II is characterized as ER-independent and it predominantly impacts postmenopausal women, as it develops from atrophic endometrium [7]. The main operative therapeutic strategy for EC is total hysterectomy with bilateral salpingo-oophorectomy in postmenopausal patients [8]. However, surgery is often combined with adjuvant chemotherapy when it comes to women with high–intermediate and high-risk ECs, as well as recurrent disease [7]. Despite current therapies, the prognosis rate for patients of the latter category remains poor. More specifically, five-year survival rates range from 23% to 72%, due to a low understanding of the molecular mechanisms behind the progression of the disease [6,7].

OC is the seventh most common gynecological malignancy, with 140,000 deaths per year reported [9,10]. In 2021, 13,770 deaths from OC were documented according to the American Cancer Society [11]. The vast majority of ovarian tumors are epithelial, which present with vague persistent gastrointestinal and urologic symptoms, such as excessive bloating and discomfort [12]. Due to the lack of screening tests for OC, patients are mostly diagnosed in the advanced stages, as determined from the tumor node metastasis classification of malignant tumors (TNM), where treatment options are severely limited [13]. More precisely, the primary approach entails surgical tumor removal combined with chemotherapy [13]. Despite all the well-developed therapies available for OC patients, OC remains the main cause of death among GC [13].

CC is the second most commonly diagnosed gynecological tumor worldwide, and thus it undoubtedly poses a major threat to women's health [14]. Each year, about 570,000 new incidents are reported, despite the massive evolution of prevention strategies during the last decades [14,15]. The main cause of CC is high-risk Human Papillomavirus (HPV) infection, which is transmitted sexually. Due to the lack of evident symptoms at early stages, many patients are often diagnosed as having advanced CC [14]. Depending on the clinical staging, radical hysterectomy with pelvic lymph node dissection and radiotherapy with or without the combination of platinum-based chemotherapy remain the standard treatments for patients with CC [14].

VGC is one of the most uncommon GCs, mostly found in postmenopausal women. Nevertheless, the growing prevalence of high-risk HPV exposure has led to an increasing incidence of VGC diagnoses [16]. Radiation therapy, particularly brachytherapy, is the preferred approach for early-stage VGC, aiming to preserve vaginal anatomy and function. However, treatment should be tailored to each patient's unique circumstances [17]. For patients with metastatic VGC, a more systematic strategy with the use of immunotherapy is being reviewed [18].

VC is one of the rarest tumors of the female genital tract, primarily impacting post-menopausal women [19]. According to the latest findings, there are two types of VC. Type I mainly afflicts younger patients and is caused by the Human Papillomavirus, resulting in vulvar intraepithelial neoplasia. Type II, conversely, predominantly manifests in elderly patients and may arise from vulvar non-neoplastic epithelial disorders, secondary to chronic inflammation [20]. Even though VC mainly presents asymptomatic, some cases are characterized by vulvar pruritus or pain, or are presented in the forms of lumps and ulcers [19]. The most common subtype of VC is the vulvar squamous cell carcinoma

(VSCC), representing approximately 5% of all gynecological malignancies [21]. The principal treatment approach involves surgical excision of the tumor, particularly for VSCC, while chemoradiation is predominantly reserved for advanced stages [19].

Although major advancements regarding the diagnosis, prognosis, and therapeutic efficacy of treatment strategies for GC have been accomplished over the past years, novel biomarkers and potential therapeutic targets are urgently needed to facilitate a more effective management of such carcinomas.

Over recent decades, a huge advancement in RNA profiling technologies has been made, indicating that there is a large amount of DNA that is transcribed but does not code for any protein. The majority of the human genome is being transcribed into non-coding RNAs, with long non-coding RNAs (lncRNAs) representing an important subclass [5]. LncRNAs are usually over 200 nucleotides long and modulate a variety of physiological processes, such as gene transcription, cell differentiation, and chromosome inactivation [15,22]. Moreover, recent evidence suggests that dysregulated lncRNAs contribute to tumorigenesis and chemoresistance [15,22]. Therefore, lncRNAs offer promising avenues as diagnostic and prognostic biomarkers, as well as targets for therapeutic intervention in diverse malignancies.

Urothelial carcinoma-associated 1 (UCA1) is a long non-coding RNA (lncRNA) with three exons that encode a 1.4 kb isoform, which acts as a tumor-enhancing gene, and a 2.2 kb isoform, characterized as the cancer upregulated drug-resistant gene in doxorubicin (DOX)-resistant epidermoid carcinoma A431 cells [22]. In recent times, a multitude of studies have been conducted to closely investigate the involvement of UCA1 in various malignancies. It was initially identified as an oncogenic lncRNA in bladder cancer, and increasing evidence indicates its pivotal role in melanoma, colorectal, gastric, and hepatocellular cancer as well [5,23]. UCA1 facilitates the reproduction of cancer cells through interacting with tumor-suppressing microRNAs (miR) and proteins, as well as signaling pathways that can regulate the post-transcriptional expression of genes involved in fundamental cell processes, such as proliferation, differentiation, and invasion [24].

Despite the accumulating evidence suggesting that overexpression of UCA1 in certain cancer entities can modulate tumorigenesis, the underlying molecular mechanisms regarding the regulatory roles of UCA1 in GC are still unknown. Consequently, it is imperative to invest efforts in understanding the purpose of UCA1 in this group of malignancies. This review aims to thoroughly examine the potential utility of UCA1 as a valuable tool in the diagnosis, prognosis, and treatment of GC.

## 2. Methods

The MEDLINE (PubMed) database was utilized for the literature review, and the data analysis consisted entirely of original research articles written in English, which focus on the role of UCA1 in GC, whereas the publications focusing on the correlation between UCA1 and other malignancies were ruled out. A total of 100 articles were identified after employing the search terms "UCA1", "breast cancer", "endometrial cancer", "ovarian cancer", "cervical cancer", "vaginal cancer", and "vulvar cancer". During the initial selection process, 25 articles were excluded, and among the 75 remaining studies, 63 relevant research articles that met the inclusion criteria were chosen for the final literature review. Figure 1 depicts the aforementioned selection process.

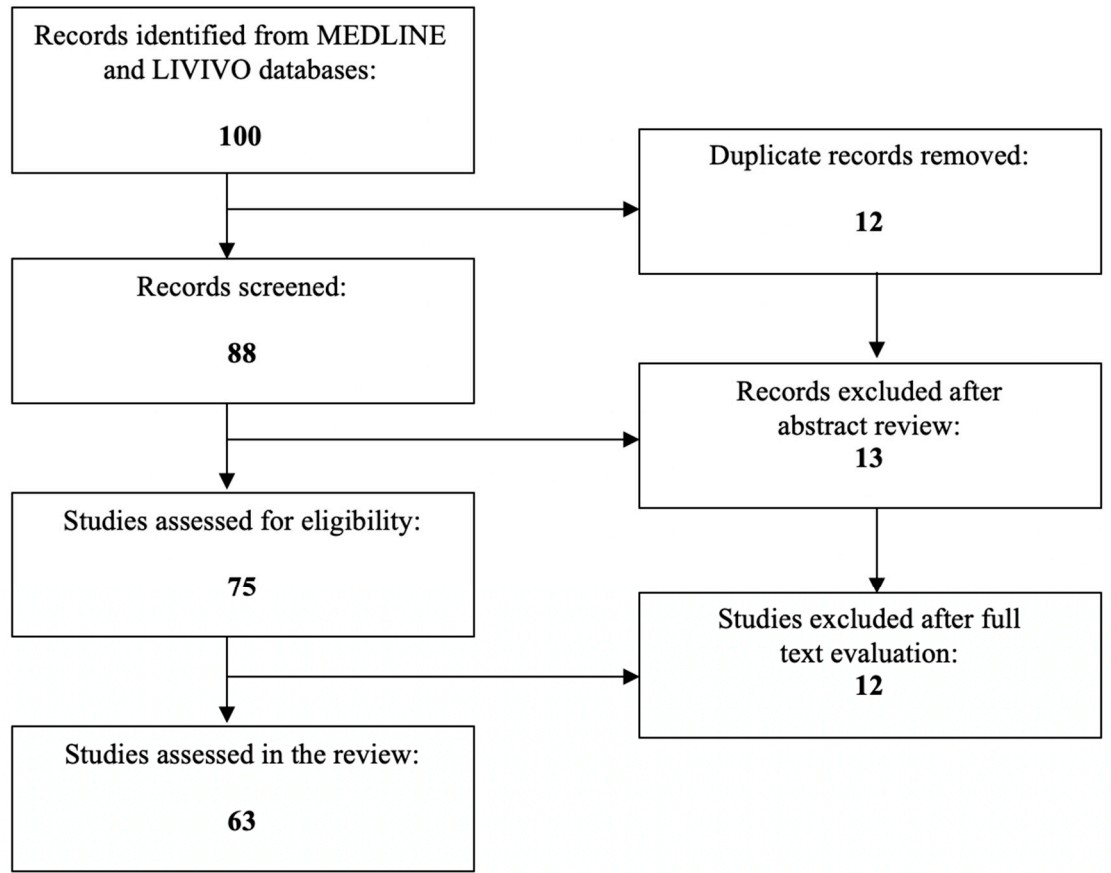

**Figure 1.** PRISMA flow diagram visually summarizing the screening process.

### 3. The Role of UCA1 in BC Oncogenesis, Proliferation, and Invasion

Numerous research groups have explored the involvement of UCA1 in BC carcinogenesis and the mechanisms behind it.

Zhou et al. observed that the downregulation of the insulin-like growth factor 2 messenger RNA-binding protein (IMP1) in the ER + PR + Her2-negative (−) or luminal A subtype T47D and MCF7 cells resulted in enhanced UCA1 expression, increasing their invasiveness. In the T47D and triple-negative MDA-MB-231 cells, this was achieved through the sponging of miR-122-5p by UCA1, and was reversed when IMP1 bound to UCA1, leading to increased pyruvate kinase M2 (PKM2) and insulin-like growth factor 1 receptor (IGF-1R) levels [24].

Furthermore, Zhang et al. found that lncRNAs UCA1 and MACC1-AS1 are mutually coordinated in MCF7 and MDA-MB-231 cells, and through sponging different miRNAs, they are able to upregulate the expression of their target mRNAs associated with oncogenic characteristics, such as increased metastasis or poor prognosis. For instance, both lncRNAs simultaneously suppressed miR-384, miR-181d-5p, and miR-10b-5p, which increased the mRNA TBL1X expression, resulting in the enhanced proliferation of MDA-MB-231 cells [25].

Moreover, Xiao and Wu discovered that the overexpression of UCA1 mediated the activation of the Wingless-related integration site (Wnt)/β-catenin pathway in MDA-MB-231 cells, leading to enhanced Epithelial–Mesenchymal Transition (EMT), which is a key factor for metastasis. Additionally, the silencing of UCA1 reduced the cells' invasion, upregulated the transmembrane protein E-cadherin, and downregulated N-cadherin, Vimentin, and the transcription factor Snail [22].

Additionally, Mota et al. found that UCA1 was downregulated in the triple-negative SUM159 cells that re-express the tumor suppressor protein Merlin but upregulated in the Merlin-deficient MCF7 and triple-negative MCF10AT cells. Its overexpression in the latter

increased hexokinase 2 (HK2), inducing aerobic glycolysis, and promoted the phosphorylation of protein kinase B (AKT) and signal transducer and activator of transcription 3 (STAT3), enhancing the cells' proliferation and decreasing their apoptosis. The transcription effectors Hippo and transforming growth factor-β (TGF-β) upregulated the UCA1 expression, further enhancing these effects [26].

Lee at al. discovered that the depletion of special AT-rich sequence-binding protein 1 (SATB1), which accounts for aggressive BC progression, increased UCA1 levels, resulting in the enhanced proliferation of MDA-MB-231 cells, while the simultaneous silencing of SATB1 and UCA1 suppressed their survival. UCA1 overexpression was also associated with increased H3K4 trimethylation (H3K4me3) and decreased H3K27 trimethylation (H3K27me3) levels, which are involved in the epigenetic modification of the lysins in Histone H3 protein and, thus, regulate gene expression [27].

In addition, Lee et al. studied the expression levels of various lncRNAs, including UCA1, in multiple BC cell lines. The results indicated that UCA1 was downregulated in the MDA-MB-231, MCF7, T47D, and Her2+ SKBR3 cells, but upregulated in the normal epithelial MCF10A cells [28].

The study of Hiemer et al. revealed that TGF-β and the transcriptional co-activator with PDZ-binding motif (TAZ) and Yes-associated protein (YAP) cooperatively upregulate UCA1 in MCF7 cells and the metastatic MDA-MB-231-derived LM2-4 cell line, increasing their oncogenic activity. UCA1 knockdown, on the other hand, decreased the migration of the TGF-β-treated LM2-4 cells [29].

Alkhathami et al. demonstrated the significant upregulation of UCA1 in the serum of untreated invasive ductal carcinoma (IDC) patients compared to healthy controls, with a remarkable increase in advanced and distant organ metastatic disease patients compared to those with early-stage disease [5].

After evaluating the expression of multiple lncRNAs in blood samples of untreated BC patients, Pourramezan et al. identified that UCA1 was significantly upregulated in BC patients compared to healthy women; however, the correlation between UCA1 levels and clinicopathological characteristics, such as race, histological grade, tumor size, TNM staging, and molecular subtypes, was not statistically significant [30].

The study of Liu et al. revealed that UCA1 levels were substantially more upregulated in the plasma of TNBC patients than the non-TNBC (NTNBC) patients, especially in those with lymph node metastasis at the time of diagnosis, which provides substantial evidence that UCA1 can be used as a TNBC-specific diagnostic biomarker [4].

Of note, El-Helkan et al. found the UCA1 expression to be markedly upregulated in the plasma of non-metastatic BC (NMBC) patients, while also discovering the considerable downregulation of UCA1 in left breast tumors of metastatic BC patients (MBC), indicating its significant association with laterality in MBC [2].

In 2017, Jiang et al. attempted to discover the specific genetic variants within multiple lncRNAs that can possibly affect the susceptibility to BC in Chinese women. UCA1 was identified as one of the significantly differentially expressed lncRNAs between the five paired tumor and normal tissues [31].

Guo et al. evaluated the UCA1 expression to be upregulated and enhance tumor development in the established transcriptional tumor suppressor ARID1A-depleted JIMT1 (Her2+), HCC1937, MDA-MB-468 (both triple-negative), and MCF7 cells, whereas the inhibition of UCA1 by ARID1A suppressed their proliferation and invasion abilities. Moreover, the ARID1A knockdown decreased the binding of CCAAT enhancer binding protein alpha gene (CEBPa) to UCA1, while ARID1A and CEBPa cooperatively downregulated UCA1 and limited the proliferation and migration of MCF7 and MDA-MB-468 cells. UCA1 overexpression abolished the tumor-suppressing effects of ARID1A in vivo, confirming the in vitro results. Lastly, although not statistically significant, the increased UCA1 levels were associated with poor prognosis in BC tissues [32].

Furthermore, Li et al. determined that UCA1 increases tumor growth in MDA-MB-231 cells by upregulating the protein tyrosine phosphatase 1B (PTP1B) expression, and the same effect

was facilitated via the inhibition of miR-206 by UCA1 in MCF7 cells. All these results were evident in vivo as well, while the analyzed tumor specimens exhibited an increase of PTP1B expression and its positive correlation with UCA1 [33].

Moreover, Guo-Yin Li et al. discovered that through the upregulation of the Suppressor of Mothers against Decapentaplegic protein 3 (SMAD3) and extracellular signal-regulated kinase (ERK), TGF-β was able to increase the expression of lncRNAs UCA1 and AC026904.1 in MCF7 and MDA-MB-231 cells. These lncRNAs cooperated to upregulate the Snail family transcriptional repressor 2 gene (Slug) and promote EMT in the cells through the inhibition of miR-1 and miR-203. They were also found more overexpressed in the triple-negative MDA-MB-231, MDA-MB-436, and BT549 (IDC-type) cells than in the MCF7, T47D, and triple-positive BT474 cells. UCA1 was able to promote invasion and metastasis in vivo, as well. The analysis of BC tissues additionally showed the upregulation of Slug by both lncRNAs, as well as the overexpression of UCA1 in IDC and metastatic specimens, its association with poor prognosis, and its ability to promote metastasis by downregulating E-cahderin [34].

Zhao et al. found UCA1 to be significantly increased in MCF7, MDA-MB-231, and especially T47D cells. UCA1 knockdown in the latter triggered the expression of the Methyltransferase 14, N6-Adenosine-Methyltransferase Subunit (METTL14), leading to the upregulation of miR-375 and the downregulation of the transcription factor SRY (sex-determining region Y)-box 2 (SOX12), and finally suppressing the proliferation–migration of the T47D cells and increasing their apoptosis. The tumor-suppressing effects from the UCA1 inhibition and its association with the METTL14-miR-375-SOX12 axis were evident in vivo, as well. Lastly, the SOX12 and UCA1 expressions were measured higher in the BC tissues compared to the normal ones, while UCA1 overexpression was associated with short overall survival [35].

Additionally, Yin et al. found UCA1 to be more upregulated in the established Natural Killer (NK)-resistant MDA-MB-231 and MCF7 cells compared to the parental lines, suggesting UCA1's ability to enhance NK resistance. UCA1 also upregulated the UL16 binding protein 2 (ULBP2), and by sponging miR-26b-5p, increased the disintegrin and metalloproteinase 17 (ADAM17) expression, which accelerated the shedding of ULBP2 from the surface of the cells and, therefore, promoted their resistance to NK. Lastly, the UCA1, ADAM17, and ULBP2 expressions were calculated higher in bone metastasis tissues of BC compared to the primary BC tissues [36].

Zhang et al. identified that the inhibition of miR-185-5p by UCA1 results in the increased proliferation and suppressed apoptosis of MCF7 cells, while the UCA1 knockdown induces the opposite outcomes. The negative correlation of UCA1 and miR-185-5p was also proven in the analysis of BC tissues, which also demonstrated upregulated UCA1 and downregulated miR-185-5p levels [37].

The study that Tuo et al. conducted revealed that UCA1 can increase the proliferation and reduce the apoptosis of MDA-MB-231 cells through the sponging of miR-143. The evaluation of BC specimens additionally showed the inverse expression between them [38].

Zhao et al. found UCA1 to be downregulated in the MCF7 cell line and to regulate the tumor necrosis factor (TNF) pathway, by controlling the expression of chemokine ligand 6 (CXCL6) and mitogen-activated protein kinase 8 (MAP3K8), that act as mediators of infection. UCA1 expression was estimated to be suppressed in the luminal BC tissues, as well, and low UCA1 levels were identified as a biomarker of poor overall survival [39].

In addition, Choudhry et al. exposed MCF7 cells to hypoxic conditions, and found that the hypoxia-inducible transcription factor 1α (HIF-1α) can upregulate the UCA1 expression and subsequently enhance the cells' survival. UCA1 was also overexpressed in the hypoxia-grown SKBR3, MDA-MB-468, MDA-MB-231, and BT474 cells, while the T47D and triple-negative BT-20 cell lines exhibited a lower increase in UCA1. Finally, the tissue analysis demonstrated that BC specimens expressed UCA1 in higher levels compared to the healthy tissues [40].

Chen et al. studied the correlation between UCA1 and macrophage infiltration, by exposing MCF7 and T47D cells to conditioned medium (CM) containing cultured human leukemia monocytic (THP-1) macrophages. The results indicated that through the activation of AKT signaling, the macrophage CM was able to upregulate the UCA1 expression and, therefore, induce significantly increased invasion abilities in the cells. The THP-1 CM-infiltrated BT474 cells overexpressed UCA1, as well. Furthermore, UCA1 levels were remarkably upregulated in BC tissues, and UCA1 overexpression was associated with advanced BC stages [41].

Finally, Záveský et al. noticed that the metastatic MDA-MB-231-derived 231BoM-1833 and 231BrM-2a variants exhibited considerably upregulated UCA1 expression, while the same was observed for the no-special-type (NST) invasive BC tissues. Notably, significantly elevated UCA1 expression in the tissues was correlated with multifocality, whereas lymph node metastasis and high UCA1 levels were slightly associated [42].

Table 1 briefly summarizes the aforementioned findings.

**Table 1.** The role of UCA1 in BB oncogenesis, proliferation, and invasion.

| Study | Study Model | Main Results |
|---|---|---|
| Zhou et al. [24] | • MDA-MB-231, T47D, and MCF7 cells | • Overexpression of UCA1 by IMP1 knockdown increases invasiveness of all cells<br>• UCA1 sponges miR-122-5p, downregulates PKM-2 and IGF-1R, and increases the invasion of MDA-MB-231 and T47D cells<br>• IMP1 binding to UCA1 reverses its sponge effect on miR-122-5p in MDA-MB-231 and T47D cells |
| Zhang et al. [25] | • MD-MB-231 and MCF7 cells | • UCA1 and MACC1-AS1 expression is mutually coordinated in both cells<br>• UCA1 and MACC1-ASI sponge miR-384, miR-181d-5p, and miR-10b-5p, upregulate TBL1X, and increase the proliferation of MDA-MB-231 cells |
| Xiao and Wu [22] | • MDA-MB-231 cells | • UCA1 activates the Wnt/beta-catenin pathway and increases EMT occurrence<br>• UCA1 knockdown reduces cells' invasion, increases E-cadherin, and decreases N-cadherin, Vimentin, and Snail |
| Mota et al. [26] | • MCF7, MCF10AT, and SUM159 cells | • Downregulation of UCA1 in SUM159 cells that re-express Merlin<br>• Overexpression of UCA1 in Merlin-deficient MCF7 and MCF10AT cells<br>• UCA1 increases HK2, promotes the phosphorylation of AKT and STAT3, and increases proliferation/reduces apoptosis<br>• Upregulation of UCA1 by Hippo and TGF-β |
| Lee et al. [27] | • MDA-MB-231 cells | • Upregulation of UCA1 by SATB1 depletion promotes the cells' survival<br>• Downregulation of SATB1 and UCA1 suppresses cells' growth<br>• UCA1 increases H3K4me3 and decreases H3K27me3 |
| Lee et al. [28] | • SKBR3, MDA-MB-231, MCF7, T47D, and MCF10A cells | • Downregulation of UCA1 in SKBR3, MDA-MB-231, MCF7, and T47D cells<br>• Upregulation of UCA1 in MCF10A cells |
| Hiemer et al. [29] | • LM2-4 and MCF-7 cells | • Upregulation of UCA1 by TGF-β, TAZ, and YAP increases the oncogenic activities of both cells<br>• UCA1 knockdown decreased the migration of LM2-4 cells |
| Alkhathami et al. [5] | • Serum of 100 untreated IDC patients and serum of 100 healthy women | • Increased UCA1 in all 100 IDC serums<br>• Increased UCA1 in advanced stages<br>• Increased UCA1 in distant organ metastatic disease |
| Pourramezan et al. [30] | • 30 whole blood samples of untreated BC patients and 30 whole blood samples of healthy women | • Upregulation of UCA1 in BC samples<br>• Association of UCA1 and clinicopathological features is not statistically significant |
| Liu et al. [4] | • 25 plasma samples and tissues of TNBC patients<br>• 35 plasma samples and tissues of NTNBC patients<br>• 40 plasma samples of healthy individuals | • Upregulation of UCA1 in TNBC samples compared to NTNBC samples<br>• Overexpression of UCA1 in lymph node metastasis |
| El-Helkan et al. [2] | • 28 MBC plasma samples<br>• 23 NMBC plasma samples<br>• 24 BB plasma samples<br>• 25 healthy plasma samples | • Upregulation of UCA1 in NMBC<br>• Downregulation of UCA1 in left MBC tumors<br>• UCA1 significantly associated with laterality in MBC |
| Jiang et al. [31] | • 5 BC tissues and 5 normal tissues | • UCA is significantly differentially expressed between BC tissues and normal tissues |

**Table 1.** *Cont.*

| Study | Study Model | Main Results |
|-------|-------------|--------------|
| Guo et al. [32] | • JIMT1, HCC1937, MDA-MB-468, MCF7, and MCF10A cells | • Overexpression of UCA1 in all ARID1A-depleted cells<br>• Downregulation of UCA1 by ARID1A decreases proliferation and migration of all cell lines<br>• ARID1A and CEBPa collaboratively downregulate UCA1 and limit the proliferation and migration of MCF7 and MDA-MB-468 cells |
| | • Female nude mice | • Overexpression of UCA1 reverses the tumor-suppressing effects of ARID1A |
| | • BC tissues and normal tissues | • Overexpression of UCA1 is associated with poor survival |
| Li et al. [33] | • MCF7 and MDA-MB-231 cells | • UCA1 upregulates PTP1B and increases tumor growth in both cells<br>• UCA1 inhibits miR-206 and upregulates PTP1B in MCF7 cells |
| | • BALB/C nude mice | • UCA1 upregulates PTP1B and increases tumor growth |
| | • 35 BC specimens and normal tissues | • UCA1 and PTP1B are positively correlated |
| Guo Yin-Li et al. [34] | • MCF7, MDA-MB-231, MDA-MB-436, BT549, T47D, and BT474 cells | • Upregulation of UCA1 by TGF-β/SMAD3 and TGF-β/ERK in MCF7 and MDA-MB-231 cells<br>• Higher UCA1 levels in MDA-MB-231, MDA-MB-436, and BT540 cells compared to MCF7, T47D, and BT474 cells<br>• UCA1 cooperates with AC026904.1 to downregulate miR-1 and miR-203, upregulate Slug, and promote EMT in MDA-MB-231 and MCF7 cells |
| | • Nude mice | • UCA1 promotes migration and metastasis |
| | • DCIS tissues<br>• IDC tissues<br>• Normal tissues | • Overexpression of UCA1 in IDC and metastatic tissues<br>• UCA1 downregulates E-cadherin and promotes metastasis<br>• UCA1 and AC026904.1 upregulate Slug in BC tissues |
| Zhao et al. [35] | • MCF7, MDA-MB-231, T47D, and MCF-10A cells | • Upregulation of UCA1 in MCF7, MDA-MB-231, and especially T47D cells<br>• Silencing of UCA1 upregulated METTL14 and miR-375, downregulated SOX12, decreased proliferation–invasion, and increased apoptosis in T47D cells |
| | • 24 nude mice | • UCA1 knockdown inhibited tumor growth via the METTL14-miR-375-SOX12 axis |
| | • 67 BC specimens and normal tissues | • Overexpression of UCA1 and SOX12 in BC tissues<br>• High UCA1 levels associated with short overall survival |
| Yin et al. [36] | • MDA-MB-231 and MCF7 cells | • Overexpression of UCA1 in NK-resistant MDA-MB-231 and MCF-7 cells promotes NK resistance<br>• UCA1 upregulates ULBP2 and ADAM17, further increasing the NK resistance in both cell lines |
| | • BC tissues<br>• Bone metastasis tissues | • Higher levels of UCA1, ADAM17, and ULBP2 in bone metastasis tissues compared to primary BC tissues |
| Zhang et al. [37] | • MCF7 cells | • UCA1 downregulates miR-185-5p, promotes tumor growth, and suppresses apoptosis |
| | • 14 BC tissues and 14 paracancer tissues | • Overexpression of UCA1 in BC tissues<br>• UCA1 and miR-185-5p are negatively correlated |
| Tuo et al. [38] | • MDA-MB-231 cells | • UCA1 sponges miR-143, promotes tumor growth, and suppresses apoptosis |
| | • 20 BC specimens and normal tissues | • UCA1 and miR-143 are negatively correlated |
| Zhao et al. [39] | • MCF7 cells | • Downregulation of UCA1<br>• UCA1 regulates CXCL6, MAP3K8, and the TNF pathway |
| | • Luminal subtype tissues | • Low UCA1 levels as a prognostic biomarker of poor survival |
| Choudhry et al. [40] | • MCF7, SKBR3, MDA-MB-468, MDA-MB-231, BT474, T47D, and BT-20 cells | • HIF-1a upregulates UCA1 and increases tumor growth in MCF7 cells<br>• Significant overexpression of UCA1 in hypoxic SKBR3, MDA-MB-468, MDA-MB-231, and BT474 cells<br>• Slight UCA1 upregulation in hypoxic T47D and BT-20 cells |
| | • 25 BC tissues and 25 normal tissues | • Overexpression of UCA1 in BC tissues |
| Chen et al. [41] | • TPH-1 CM MCF7, T47D, and BT474 cells | • UCA1 upregulation in macrophage-infiltrated MCF7, T47D, and BT4 cells<br>• Macrophage upregulates UCA1 through AKT activation and increases invasion of MCF7 and T47D cells |
| | • 71 BC tissues and 71 normal tissues | • Overexpressed UCA1 in BC tissues<br>• High UCA1 levels associated with advanced clinical stage |

**Table 1.** *Cont.*

| Study | Study Model | Main Results |
|---|---|---|
| Záveský et al. [42] | • 231BoM-1833 and 231BrM-2a cells | • Overexpression of UCA1 in both cells |
| | • 29 NST invasive BC tissues<br>• 29 benign tissues | • Overexpression of UCA1 associated with multifocality in NST invasive BC tissues<br>• Slight correlation between UCA1 and lymph node metastasis in NST invasive BC tissues |

## 4. Therapeutic Implications of UCA1 in BC

Many studies have been conducted to investigate the role of UCA1 in BC chemoresistance, and the repercussions of its interactions with various drugs and natural elements in BC treatment.

Okcanoğlu et al. found that the Aurora kinase inhibitor CCT137690 is able to downregulate UCA1 in MDA-MB-231 cells, possibly through the direct inhibition of fibroblast growth factor receptor 1 (FGFR1); however, the latter was not confirmed [43].

Furthermore, Mokhtary et al. investigated the potential use of UCA1 in BC gene therapy. They synthesized a UCA1 short-hairpin RNA (shRNA) to cause UCA1 knockdown via RNA interference (RNAi), and then established a UCA1 shRNA complex formed by a vesicular nanocarrier consisting of polysorbate 80 or Tween 80 (T), squalene (S), and cationic lipid didodecyldimethylammonium bromide (DDAB), with a cationic polymer (PEI) ((T:S)1040 µM with PEI). The UCA1 shRNA-(T:S) with PEI complex was used to treat MCF7 cells, which exhibited increased apoptosis and G2/M cell cycle arrest. This evidence suggests that UCA1 downregulation suppresses tumor growth by inducing cell cycle arrest in MCF7 cells. Accordingly, the application of UCA1 RNAi through the vesicular nanocarrier (T:S)1040 µM with PEI constitutes a promising strategy for the future of BC gene treatment [44].

Moreover, Rezaie et al. detected that UCA1 can be successfully downregulated by quercetin, a flavonoid with known anti-cancer properties, leading to G2 cell cycle arrest in MCF7 cells, suppressed proliferation, and enhanced apoptosis [45].

Zhu et al. investigated the involvement of UCA1 in trastuzumab resistance and found that the established trastuzumab-resistant SKBR3 cells exhibited higher UCA1 levels than the parental cells. The UCA1 knockdown upregulated miR-18a, and downregulated its direct target YAP1, facilitating the trastuzumab-triggered apoptosis of the cells, and significantly limiting their invasiveness [46].

The study of Jiang et al. illustrated that UCA1 was significantly differentially expressed between Adriamycin (ADR)-sensitive MCF7 and ADR-resistant MCF7 cells, suggesting of UCA1 ability to mediate ADR resistance [47].

Additionally, Wu and Luo examined the role of UCA1 in tamoxifen resistance. They found that UCA1 levels were lower in the tamoxifen-sensitive MCF7 cells compared to the MCF7-derived tamoxifen-resistant LCC2 and LCC9 cells, which overexpressed AKT and the mammalian target of rapamycin (mTOR) but exhibited decreased viability and increased apoptosis following the UCA1 knockdown. The MCF7 cells demonstrated tamoxifen resistance after being infected with UCA1 particles; however, their exposure to rapamycin abolished the protective effect of UCA1 on them. These findings suggest that UCA1 induces tamoxifen resistance by activating the AKT/mTOR pathway [48].

Li et al. also found that the tamoxifen-resistant LCC2, LCC9, and BT474 cells exhibited markedly overexpressed UCA1 compared to MCF7 cells. After treating the latter with tamoxifen, they estimated a significant upregulation of UCA1 and HIF-1a, and a decrease in miR-18a, which promoted the cells' viability. The silencing of UCA1 in LCC9 and BT474 cells had the opposite effects and enhanced their sensitivity to tamoxifen. Altogether, tamoxifen increases UCA1 in ER+ cells, leading to the inhibition of miR-18a and the upregulation of HIF-1a, all of which contribute to acquired tamoxifen resistance [49].

Besides, Xu et al. isolated the exosomes released from MCF7 and LCC2 cells and identified that LCC2 cells, and particularly their exosomes, overexpressed UCA1 compared to MCF7 cells. The latter were treated with LCC2/exosomes and exposed to tamoxifen, which increased their viability and decreased their apoptosis. On the contrary, the ability of the LCC2/exosomes with deficient UCA1 expression to promote tamoxifen resistance in MCF7 cells was significantly reduced [50].

Furthermore, Liu et al. noticed that the generated tamoxifen-resistant MCF7 and T47D cells (MCF7-R and T47D-R) exhibited higher UCA1 expression compared to their parental lines, resulting in increased proliferation and migration abilities. Their treatment with tamoxifen, however, caused the downregulation of UCA1, which decreased their survival and enhanced their sensitivity to tamoxifen through the inhibition of β-catenin. The in vivo experiment validated all the aforementioned findings. Lastly, the stage III and IV hormone receptor-positive (HR+) tissues expressed higher UCA1 and β-catenin levels than the specimens of stages I and II, while UCA1 overexpression was associated with poor prognosis [23].

In addition, Zhuo Li et al. found that LCC2 and LCC9 cells demonstrated higher UCA1 expression than the MCF7 and T47D cells. Furthermore, through inducing G2/M cycle arrest, UCA1 knockdown inhibited the phosphoinositide 3 kinase (PI3K)/AKT axis, which downregulated CAMP-responsive element binding protein (CREB), leading to increased apoptosis and tamoxifen sensitivity in LCC2 and LCC9 cells. The enforced UCA1 expression on MCF7 and T47D cells recruited the enhancer of zeste homolog 2 (EZH2), which downregulated the cyclin-dependent kinase inhibitor p21, and decreased the tamoxifen sensitivity of the cells. Finally, the overexpression of UCA1 promoted tumor progression in HR+ BC tissues [3].

Liu et al. observed that the established paclitaxel (PTX)-resistant MCF7 cells (MCF7/PTX) expressed higher UCA1 levels than the MCF7 and MCF10A cells, and UCA1 was able to promote PTX resistance by upregulating cyclin-dependent kinase 12 (CDK12) through the sponging of miR-613 in MCF7/PTX cells. The in vivo experiment showed that UCA1 increased the tumor volume and induced PTX resistance via the miR-613/CDK12 axis. Lastly, PTX-resistant tissues exhibited more upregulated UCA1 expression that the PTX-sensitive specimens [51].

Moreover, Huang et al. discovered that hnRNP I, a member of the heterogeneous nuclear ribonucleoproteins family (hnRNP), increases UCA1 stability in MCF7 and MDA-MB-231 cells, making them resistant to doxorubicin (DOX). Additionally, by competing with the cyclin-dependent kinase inhibitor protein p27, UCA1 promotes tumor development in MCF7 cells. In the in vivo experiment, UCA1 enhanced cancer proliferation, while the tissue specimens analysis showed that UCA1 promotes tumor growth by suppressing p27 [52].

Wo et al. demonstrated that the upregulation of UCA1 by TFG-β resulted in enhanced EMT and DOX resistance in MCF7, MDA-MB-231, and MDA-MB-468 cells. Moreover, BC tissues exhibited more elevated UCA1 levels compared to normal specimens [53].

Table 2 briefly summarizes the aforementioned findings.

**Table 2.** Therapeutic implications of UCA1 in BC.

| Study | Study Model | Main Results |
|---|---|---|
| Okcanoğlu et al. [43] | • MDA-MB-231 cells | • CCT137690 downregulates UCA1 |
| Mokhtary et al. [44] | • MCF7 cells | • UCA1 shRNA-(T:S)1040 µM with PEI complex increases apoptosis by inducing G2/M cell cycle arrest |
| Rezaie et al. [45] | • MCF7 cells | • Downregulation of UCA1 by quercetin results in G2 cell cycle arrest, increased apoptosis, and decreased proliferation |
| Zhu et al. [46] | • SKBR3<br>• Trastuzumab-resistant SKBR3 cells | • Overexpression of UCA1 in trastuzumab-resistant cells<br>• UCA1 knockdown increases trastuzumab-triggered apoptosis and decreases invasion through the upregulation of miR-18a and downregulation of YAP1 in trastuzumab-resistant cells |

**Table 2.** *Cont.*

| Study | Study Model | Main Results |
|---|---|---|
| Jiang et al. [47] | • MCF7 <br> • MCF7/ADR cells | • UCA1 is significantly differentially expressed between MCF7 and MCF7/ADR cells |
| Wu and Luo [48] | • MCF7, LCC2, and LCC9 cells | • Higher UCA1 levels in LCC2 and LCC9 cells compared to MCF7 cells <br> • UCA1 knockdown decreases viability and increases apoptosis of LCC2 and LCC9 cells <br> • UCA1 increases tamoxifen resistance by upregulating AKT and mTOR in LCC2 and LCC9 cells <br> • Rapamycin abrogates the protective effect of UCA1 in the UCA1-infected MCF7 cells |
| Li et al. [49] | • MCF7, LCC2, LCC9, and BT474 cells | • Higher UCA1 levels in LCC2, LCC9, and BT474 cells compared to MCF7 cells <br> • Tamoxifen treatment increases UCA1 and HIF-1a, decreases miR-18a, and enhances the viability of MCF7 cells |
| Xu et al. [50] | • MCF7 cells and exosomes, LCC2 cells and exosomes | • Higher UCA1 levels in LCC2/exosomes compared to MCF7/exosomes <br> • Increased viability and decreased apoptosis of MCF7 cells treated with LCC2/exosomes after tamoxifen treatment <br> • Reduced ability of LCC2/exosomes with impaired UCA1 to promote tamoxifen resistance to MCF7 cells |
| Liu et al. [23] | • MCF7-R and T47D-R cells | • Overexpression of UCA1 increases the proliferation and migration of both cells <br> • Tamoxifen treatment downregulates UCA1, decreases survival, and enhances tamoxifen sensitivity through the inhibition of β-catenin in both cells |
| | • Mice xenograft | • UCA1 silencing decreases tumor growth and size <br> • UCA1 knockdown increases tamoxifen sensitivity |
| | • 30 HR+ BC specimens, stages I and II <br> • 24 HR+ BC specimens, stages III and IV <br> • 14 normal tissues | • Overexpression of UCA1 and β-catenin in stages III and IV <br> • UCA1 promotes tamoxifen resistance <br> • High UCA1 levels associated with poor survival |
| Zhuo Li et al. [3] | • MCF7, T47D, LCC2, and LCC9 cells | • Higher UCA1 levels in LCC2 and LCC9 cells compared to MCF7 and T47D cells <br> • UCA1 knockdown induces G2/M cycle arrest, inhibits the PI3K/AKT axis, downregulates CREB, and increases apoptosis and tamoxifen sensitivity in LCC2 and LCC9 cells <br> • The recruitment of EZH2 by UCA1 downregulates p21 and decreases tamoxifen sensitivity of MCF7 and T47D cells |
| | • 10 HR+ BC tissues and 10 normal tissues | • Overexpression of UCA1 promotes tumor growth in BC tissues |
| Liu et al. [51] | • MCF7, MCF7/PTX, and MCF10A cells | • Higher UCA1 levels in MCF7/PTX cells compared to MCF7 and MCF10A cells <br> • UCA1 induces PTX resistance by sponging miR-613 and upregulating CDK12 in MCF7/PTX cells |
| | • 24 BALB/c nude mice | • UCA1 increases tumor volume, downregulates miR-163, upregulates CDK12, and promotes PTX resistance |
| | • 30 PTX-resistant BC tissues <br> • 30 PTX-sensitive BC tissues | • Overexpression of UCA1 in PTX-resistant tissues |
| Huang et al. [52] | • MCF7 and MDA-MB-231 cells | • hnRNP 1 enhances UCA1 stability and induces DOX resistance in both cells <br> • UCA1 competes with p27 for hnRNP I and increases tumor growth in MCF7 cells |
| | • Female nude mice | • UCA1 increases cancer proliferation |
| | • BC specimens and normal specimens | • UCA1 downregulates p27 and increases tumor growth |
| Wo et al. [53] | • MCF7, MDA-MB-231, and MDA-MB-468 cells | • TGF-β upregulates UCA1 and induces EMT and DOX resistance in all cells |
| | • 15 BC tissues and 15 healthy tissues | • Overexpression of UCA1 in BC tissues |

## 5. The Role of UCA1 in EC Oncogenesis, Proliferation, and Invasion

Two study groups have explored the contribution of UCA1 in EC development.

Liu et al. generated three primary EC cell lines derived from endometrioid adenocarcinoma patients and found that UCA1 enhances their proliferation by sponging miR-143-3p

and upregulating Kruppel-like factor 5 (KLF5) and promotes EMT through downregulating miR-1-3p and increasing the relaxin-like family peptide receptor 1 (RXFP1), respectively. The in vivo experiment revealed that UCA1 silencing suppressed cancer development, while UCA1 was overexpressed in the endometrioid EC tissues and closely correlated with tumor growth, metastasis, and poor overall survival [8].

Lu et al. demonstrated that UCA1 silencing decreased the migration and invasion of type II adenosquamous carcinoma HTB-111 and type I endometrioid adenocarcinoma Ishikawa cells. Furthermore, the adenocarcinoma and non-adenocarcinoma tissue types I and II, predominantly the lymph node metastasis specimens, exhibited considerably elevated UCA1 expression. Upregulated UCA1 levels were associated with distant metastasis, advanced stage, high histological grade, and poor prognosis [6].

Table 3 briefly summarizes the aforementioned findings.

**Table 3.** The role of UCA1 in EC oncogenesis, proliferation, and invasion.

| Study | Study Model | Main Results |
|---|---|---|
| Liu et al. [8] | • 3 Primary patient-derived endometrioid EC cell lines | • Overexpression of UCA1<br>• UCA1 promotes cells' proliferation and survival by sponging miR-143-3p and upregulating KLF5<br>• UCA1 downregulates miR-1-3-p, increases RXFP1 expression, and promotes EMT |
| | • Female BALB/c nude mice | • UCA1 knockdown suppresses tumor growth |
| | • 64 endometrioid adenocarcinoma specimens and 64 normal tissues | • Overexpression of UCA1 in EC tissues<br>• UCA1 associated with EC progression and metastasis<br>• High UCA1 levels associated with poor survival |
| Lu et al. [6] | • HTB-111 and Ishikawa cells | • UCA1 knockdown decreases migration and invasion of both cell lines |
| | • 15 proliferative endometrium samples<br>• 45 EC tissues<br>• 15 lymph node metastasis tissues of EC | • Overexpression of UCA1 in EC tissues<br>• Highest UCA1 levels in the lymph node metastasis tissues<br>• High UCA1 levels associated with lymph node and distant metastasis, advanced stage, high histological grade, and poor prognosis |

## 6. Therapeutic Implications of UCA1 in EC

The study that Dong et al. conducted revealed that the invasive, sphere-forming, and PTX-resistant derivatives of the poorly differentiated endometrioid type II HEC-50 cells exhibited significantly more overexpressed UCA1 levels compared to their parental cell line [54].

Table 4 briefly summarizes the aforementioned findings.

**Table 4.** Therapeutic implications of UCA1 in EC.

| Study | Study Model | Main Results |
|---|---|---|
| Dong et al. [54] | • HEC-50 cells<br>• HEC-50 invasive, sphere-forming, and PTX-resistant derivatives | • Overexpression of UCA1 in the HEC-50 invasive, sphere-forming, and PTX-resistant derivatives compared to the HEC-50 cells |

## 7. The Role of UCA1 in OC Oncogenesis, Proliferation, and Invasion

Several studies have investigated the contribution of UCA1 to OC pathogenesis.

Liu et al. demonstrated the considerable upregulation of UCA1 in the metastatic SKOV3.ip1 cells compared to their parental serous cystadenocarcinoma SKOV3 line, which indicates the heightened potential of UCA1 to promote metastasis [55].

Furthermore, Qiu et al. additionally showed the correlation of UCA1 with metastasis in OC. They observed that OC tissues exhibited more elevated UCA1 expression than the

benign and normal ovarian specimens, while this overexpression was significantly associated with some clinicopathological characteristics, including staging, grade, peritoneal effusion, and lymph node metastasis [56].

Moreover, Lin et al. found that UCA1 increased YAP expression after binding to its regulator, Angiomotin (AMOT), enhancing the interaction between YAP and AMOT, and therefore mediating the YAP dephosphorylation and nuclear translocation. This resulted in the increased survival and proliferation of high-grade serous adenocarcinoma CaOV3 and hereditary BC-OC syndrome (BRCA1)-associated UWB1.289 cells, serous cystadenocarcinoma OVCA429 cells, and ovarian surface epithelial OSEC4C2 cells. Moreover, silencing of UCA1 led to significant tumor suppression in vivo. UCA1 was overexpressed in high-grade serous adenocarcinoma tissues and significantly associated with prognosis, while its locus was marked by a tumor-specific super-enhancer, regulating its expression. Treatment with the inhibitor of the bromodomain and extra-terminal domain (BET) family of proteins (+)-JQ1, however, resulted in the downregulation of UCA1 in CaOV3 and UWB1.289 cells, confirming the research suggesting that super-enhancer-associated genes are sensitive to (+)-JQ1 [57].

The study of Xu et al. indicated that through sponging miR-99b-3p, UCA1 was able to regulate the expression of serine/arginine-rich splicing factor protein kinase-1 (SRPK1) and increase the viability of OC cells, which overexpressed UCA1. The examined OC tissues exhibited upregulation of UCA1, as well [58].

Additionally, Yang et al. showed the UCA1 overexpression in SKOV3, and particularly in mucinous cystadenocarcinoma OMC685 and endometrioid adenocarcinoma A2780 cells, in comparison to the normal ovarian IOSE386 cells. UCA1 was able to promote the migration and invasion of the last two OC cells by inhibiting miR-485-5p and subsequently upregulating matrix metallopeptidase 14 (MMP14). The positive correlation of UCA1 and MMP14 was additionally demonstrated in the epithelial OC tissues (EOC), where UCA1 levels were increased and associated with the International Federation of Gynecology and Obstetrics (FIGO) stage, lymph node metastasis, and poor prognosis [59].

Table 5 briefly summarizes the aforementioned findings.

**Table 5.** The role of UCA1 in OC oncogenesis, proliferation, and invasion.

| Study | Study Model | Main Results |
|---|---|---|
| Liu et al. [55] | • SKOV3 <br> • SKOV3.ip1 cells | • Upregulation of UCA1 in SKOV3.ip1 cells |
| Qiu et al. [56] | • 26 OC tissues <br> • 16 normal and benign ovarian tissues | • Overexpression of UCA1 in OC tissues <br> • High UCA1 levels associated with staging, grade, peritoneal effusion, and lymph node metastasis |
| Lin et al. [57] | • CaOV3, UWB1.289, OVCA429, and OSEC4C2 cells | • UCA1 promotes YAP–AMOT interaction, mediates the YAP dephosphorylation and nuclear translocation, and increases survival and proliferation of all cells <br> • Downregulation of UCA1 in (+)-JQ1-treated CaOV3 and UWB1.289 cells |
| | • Female nu/nu mice | • Depletion of UCA1 suppresses tumor growth |
| | • High-grade serous adenocarcinoma tissues | • Overexpression of UCA1 <br> • High UCA1 levels associated with prognosis <br> • A tumor-specific super-enhancer regulates UCA1 in its locus |
| Xu et al. [58] | • OC cells | • UCA1 overexpression sponges miR-99b-3p, regulates SRPK1, and increases cells' viability |
| | • OC tissues | • Upregulation of UCA1 |
| Yang et al. [59] | • SKOV3, OMC685, A2780, and IOSE386 cells | • Overexpression of UCA1 in SKOV3, and especially OMC685 and A2780 cells <br> • UCA1 sponges miR-485-5p, upregulates MMP14, and increases migration and invasion of OMC685 and A2780 cells |
| | • 53 EOC tissues <br> • 29 normal tissues | • UCA1 increases MMP14 <br> • High UCA1 levels associated with FIGO stage, lymph node metastasis, and poor survival |

## 8. Therapeutic Implications of UCA1 in OC

The role of UCA1 in OC chemoresistance was the subject of interest for multiple researchers.

The study of Wang et al. revealed the significant upregulation of UCA1 in the generated PTX-resistant SKOV3 (SKOV3/PTX) and high-grade serous adenocarcinoma HeyA8 (HeyA8/PTX) cells compared to their parental lines. Inhibition of UCA1 in the PTX-resistant cells, however, led to the upregulation of miR-129 and the following depletion of ATP binding cassette subfamily B member 1 (ABCB1), which promoted the PTX-induced apoptosis of the cells [60].

Horita et al. found that both UCA1 and Oncolytic Vaccinia Virus (OVV) expressions were upregulated in the serous cystadenocarcinoma PTX-resistant KFTX and KFTXlow cells compared to PTX-sensitive KFlow cells, indicative of UCA1's ability to promote both PTX resistance and viral replication. Moreover, SKOV3 and clear cell adenocarcinoma RMG-1 cells exhibited higher UCA1 and enhanced green fluorescent protein (EGFP) levels, and more cytopathic effects and OVV replication than serous cystadenocarcinoma SHIN3 and high-grade serous adenocarcinoma ES-2 and OVCAR3 cells. Therefore, UCA1 can control the oncolytic properties of OVV, and potentially serve as a valuable biomarker for predicting its efficacy in OC. Additionally, by triggering the Cell Division Cycle 42 (Cdc42) expression, UCA1 enhanced the cell-to-cell spread of OVV in SKOV3 cells. The in vivo experiment also illustrated that OVV treatment should be preferred over PTX for tumors overexpressing UCA1, and vice versa [61].

In addition, Li et al. demonstrated the remarkable UCA1 overexpression in A2780, serous cystadenocarcinoma OAW42, high-grade serous adenocarcinoma OVCAR4, and especially SKOV3 and HeyA8 cells, compared to IOSE-386 cells. Notably, UCA1 was estimated as even more overexpressed in the established SKOV3/PTX and HeyA8/PTX cells, which exhibited a substantial decrease in their proliferation, migration, and invasion, as well as an increase in their apoptosis following the inhibition of UCA1. UCA1 was additionally able to promote PTX resistance in these cells via simultaneously downregulating miR-654-5p and upregulating Salt-Inducible Kinase 2 (SIK2). Lastly, OC tissues showed higher UCA1 and lower miR-654-5p levels than the adjacent normal specimens [62].

Zhang et al. found that UCA1 was overexpressed in OEC tissues and significantly correlated with advanced FIGO stage, lymph node metastasis, short survival rates, and resistance to chemotherapy in OEC tissues. Additionally, even though the cisplatin (DDP)-resistant patients exhibited higher UCA1 expression than the DDP-sensitive group, this disparity was not statistically significant [63].

The study of Li et al. indicated that the established DDP-resistant A2780 (A2780/DDP) and SKOV3 (SKOV3/DDP) cells overexpressed UCA1 in comparison to their parental lines and the normal epithelial IOSE-80 cells. Nonetheless, a remarkable decrease in their proliferation and an increase in their DDP-induced apoptosis was observed following UCA1 knockdown. Moreover, the inhibition of miR-143 and subsequent upregulation of FOS-like 2 AP-1 transcription factor subunit (FOSL2) by UCA1 facilitated its effects on DDP resistance in the cells. UCA1 enhanced cell growth and reduced the DDP sensitivity in vivo, as well. The DDP-sensitive serous OC tissues showed a higher UCA1 upregulation than the normal ones; however, the highest UCA1 levels were detected in the DDP-resistant specimens, which additionally exhibited suppressed miR-143 and heightened FOSL2 levels, respectively. Lastly, the DDP-resistant serum-derived exosomes demonstrated UCA1 overexpression and its negative association with miR-143 [64].

Besides, Wambecke et al. revealed that UCA1 was responsible for enhancing DDP resistance to both DDP-resistant OAW42 (OAW42/DDP) cells and their parental line, whereas UCA1 silencing induced a S-G2/M phase block in the former cells and increased apoptosis at G1 phase in DDP-sensitive OAW42 and high-grade serous adenocarcinoma OVCAR3 cells. Additionally, suppression of the short isoform of UCA1 significantly increased the DDP sensitivity to OAW42/DDP cells through the upregulation of miR-27a-5p and the following downregulation of ubiquitin-conjugating enzyme E2 N (UBE2N), which increased B-cell lymphoma 2 (Bcl2)-like protein 11 (BIM). Finally, OC tissues with high UCA1 levels presented a significantly shorter median progression-free survival (PFS) compared to the tissues with a lower UCA1 expression [65].

Wang et al. showcased the ability of UCA1 to promote the proliferation, migration, invasion, and resistance of SKOV3 cells to DDP, through the upregulation of SRPK1, and the subsequent overexpression of the antiapoptotic Bcl2 protein and the downregulation of the proapoptotic Bcl-2-Associated X-protein (Bax), caspase-3, and caspase-9 levels. Furthermore, the EOC tissues displayed markedly overexpressed UCA1 and SRPK1 levels compared to the normal specimens [66].

Table 6 briefly summarizes the aforementioned findings.

**Table 6.** Therapeutic implications of UCA1 in OC.

| Study | Study Model | Main Results |
|---|---|---|
| Wang et al. [60] | • SKOV3 and SKOV3/PTX cells <br> • HeyA8 and HeyA8/PTX cells | • Overexpression of UCA1 in SKOV3/PTX and HeyA8/PTX cells <br> • UCA1 knockdown increases miR-129, downregulates ABCB1, and enhances PTX-induced apoptosis of SKOV3/PTX and HeyA8/PTX cells |
| Horita et al. [61] | • KFTX, KFTXlow, KFlow, SKOV3, RMG-1, SHIN3, ES-2, and OVCAR3 cells | • Higher UCA1 and OVV expressions in KFTX and KFTXlow cells compared to KFlow cells <br> • Higher UCA1 and EGFP levels, and more cytopathic effects and OVV replication in SKOV3 and RMG-1 cells than in SHIN3, ES-2, and OVCAR3 cells <br> • UCA1 enhances cell-to-cell spread of OVV by activating Cdc42 in SKOV3 cells |
| | • Female BALB/cAJcl-nu/nu mice | • OC with overexpressed UCA1 is more sensitive to OVV than PTX, and vice versa |
| Li et al. [62] | • A2780, OAW42, OVCAR4, SKOV, HeyA8, SKOV3/PTX, HeyA8/PTX, and IOSE-386 cells | • Overexpression of UCA1 in all cells, especially SKOV3 and HeyA8, except for IOSE-386 cells <br> • Highest UCA1 levels in SKOV3/PTX andHeyA8/PTX cells <br> • UCA1 knockdown suppresses the proliferation, migration, and invasion, and increases the apoptosis of SKOV3/PTX andHeyA8/PTX cells <br> • UCA1 induces PTX resistance by sponging miR-654-5p and upregulating SIK2 in SKOV3/PTX andHeyA8/PTX cells |
| | • 31 OC tissues and normal tissues | • Upregulation of UCA1 and downregulation of miR-654-5p in OC tissues |
| Zhang et al. [63] | • 117 DDP-resistant and DDP-sensitive EOC tissues <br> • Normal tissues | • UCA1 overexpression associated with advanced FIGO stage, lymph node metastasis, poor prognosis, and resistance to chemotherapy in OEC tissues <br> • Higher UCA1 levels in DDP-resistant tissues than in DDP-sensitive tissues, but not statistically significant |
| Li et al. [64] | • A2780 and A2780/DDP cells <br> • SKOV3 and SKOV3/DDP cells <br> • IOSE-80 cells | • Overexpression of UCA1 in A2780/DDP and SKOV3/DDP cells <br> • UCA1 knockdown decreases proliferation and increases DDP-induced apoptosis in A2780/DDP and SKOV3/DDP cells <br> • UCA1 inhibits miR-143 and upregulates FOSL2 to enhance DDP resistance in A2780/DDP and SKOV3/DDP cells |
| | • 12 BALB/c female athymic mice | • UCA1 increases cell growth and decreases DDP sensitivity |
| | • 32 DDP-resistant serous OC tissues <br> • 24 DDP-sensitive serous OC tissues <br> • 56 normal tissues | • UCA1 is higher in DDP-sensitive tissues compared to normal tissues <br> • Highest UCA1 expression in DDP-resistant tissues <br> • Decreased miR-143 and increased FOSL2 in DDP-resistant tissues |
| | • 56 OC serum-derived exosomes | • UCA1 upregulation and miR-143 downregulation in DDP-resistant serum-derived exosomes |
| Wambecke et al. [65] | • OAW42 and OAW42/DDP cells <br> • OVCAR3 cells | • UCA1 enhances DDP resistance in OAW42 and OAW42/DDP cells <br> • UCA1 knockdown causes S-G2/M phase block in OAW42/DDP cells and G1 phase apoptosis in OAW42 and OVCAR3 cells <br> • Silencing of short isoform of UCA1 increases DDP resistance in OAW42/DDP cells by upregulating miR-27a-5p, downregulating UBE2N, and increasing BIM |
| | • OC tissues from cohorts GSE26193 and GSE9891 | • Shorter PFS in tissues with high UCA1 levels |
| Wang et al. [66] | • SKOV3 cells | • UCA1 enhances proliferation, migration, invasion, and DDP resistance by upregulating SRPK1 and Bcl2, and downregulating Bax, caspase-3, and caspase-9 |
| | • 24 EOC tissues <br> • 16 normal tissues | • Higher UCA1 and SRPK1 levels in EOC tissues |

**9. The Role of UCA1 in CC Oncogenesis, Proliferation, and Invasion**

Nine research groups have, to this day, uncovered the role of UCA1 in the tumorigenesis of CC.

Duan et al. discovered that UCA1 knockdown suppressed tumor progression and enhanced the apoptosis of HeLa cells (adenocarcinoma) through the downregulation of β-catenin and transcription factor 4 (TCF-4) [67].

Yan et al. found UCA1 to be upregulated in human CC cells, while its inhibition led to the overexpression of miR-206 and the suppression of the vascular endothelial growth factor (VEGF). This resulted in a significant decrease in the proliferation, migration, invasion, and viability of the cells [68].

Furthermore, Gao et al. demonstrated that the CaSki-derived exosomes exhibited overexpressed UCA1 and SRY-Box Transcription Factor 2 (SOX2) levels, but decreased miR-122-5p expression, whereas the silencing of UCA1 in the CD133+CaSki stem cells (squamous cell carcinoma or SCC) was able to upregulate miR-122-5p and downregulate SOX2, resulting in their suppressed proliferation, migration, and invasion. In vivo, UCA1 knockdown increased the apoptosis and reduced the tumor volume in the mice [69].

Moreover, He et al. found that UCA1 was overexpressed in SiHa, ME180, C33a (all SCC), CaSki, and HeLa cells, but downregulated in the epithelial non-cancerous Ect1/E6E7 cells. UCA1 promoted the proliferation and invasion of SiHa and CaSki cells through sponging miR-204 and upregulating Kinesin Family Member 20A (KIF20A). The in vivo experiment revealed significant tumor suppression following the UCA1 knockdown. Lastly, CC specimens exhibited upregulated UCA1 and low survival rates, while the inhibition of UCA1 decreased the KIF20A expression [70].

An et al. identified UCA1 to be upregulated in HeLa, SiHa, and particularly ME180 cells, leading to a significant increase in proliferation, migration, and invasion in the first two. UCA1 overexpression was shown to downregulate the SWI/SNF-related, matrix-associated, actin-dependent regulator of chromatin, subfamily d, member 3 (SMARCD3) via ubiquitin-mediated proteolysis in HeLa and ME180 cells, resulting in the progression of tumor growth. The ME180 mouse xenograft models exhibited a decrease in cancer development following the silencing of UCA1, while demonstrating a negative correlation between UCA1 and SMARCD3. Finally, UCA1 was evaluated as overexpressed in both tumor tissues and CC plasma exosomes [71].

Additionally, An et al. showed that HeLa, CaSki, and ME180 cells exhibited significant UCA1 upregulation, which enhanced their proliferation and invasion abilities. This was also accomplished in the SiHa cell line through the inhibition of miR-299-3p. The downregulation of miR-299-3p and subsequent tumor development caused by UCA1 overexpression were obvious in the CC tissues, as well [72].

Besides, Wei et al. showed that UCA1 promoted the proliferation, migration, and invasion of HeLa cells by downregulating miR-145, which was additionally evident in the CC tissues, where UCA1 accelerated the cancer progression [14].

Wu et al. identified that N3CA (SCC), RL95-2, lshikawa3H12, HEC-1A (all adenocarcinoma), and especially HeLa and HEC-1B (adenocarcinoma) cells overtly express UCA1. UCA1 seemed to promote the proliferation and glycolysis of the last two through sponging miR-493-5p, which targeted HK2. UCA1 expression was additionally examined in 20 CC specimens, revealing upregulation in 13 of them [73].

Yang et al. found that UCA1 suppressed miR-155 in order to promote EMT and enhance the proliferation, migration, and invasion of HeLa cells. The CC tissues exhibited decreased miR-155 and enhanced UCA1 expression, with the latter being significantly associated with short overall survival [74].

Table 7 briefly summarizes the aforementioned findings.

**Table 7.** The role of UCA1 in CC oncogenesis, proliferation, and invasion.

| Study | Study Model | Main Results |
|---|---|---|
| Duan et al. [67] | • HeLa cells | • UCA1 knockdown decreases proliferation and increases apoptosis by downregulating β-catenin and TCF-4 |
| Yan et al. [68] | • Human CC cells | • Overexpression of UCA1<br>• UCA1 knockdown upregulates miR-206 and downregulates VEGF, leading to decreased proliferation, migration, invasion, and cell viability |
| Gao et al. [69] | • CD133+CaSki stem cells<br>• CaSki-derived exosomes | • Overexpression of UCA1 and SOX2, and downregulation of miR-122-5p in CaSki-exosomes<br>• UCA1 knockdown decreases SOX2, increases miR-122-5p, and reduces proliferation, migration, and invasion in CD133+CaSki stem cells |
| | • 60 female nude mice | • UCA1 knockdown increases apoptosis and suppresses tumor volume |
| He et al. [70] | • SiHa, HeLa, ME180, C33a, CaSki, and Ect1/E6E7 cells | • Overexpressed UCA1 in SiHa, HeLa, ME180, C33a, and CaSki cells<br>• Suppressed UCA1 in Ect1/E6E7<br>• UCA1 enhances proliferation and invasion by sponging miR-204 and upregulating KIF20A in SiHa and CaSki cells |
| | • Female nude mice | • UCA1 knockdown suppresses tumor growth |
| | • 8 CC tissues and normal tissues | • Overexpression of UCA1 associated with poor survival in CC tissues<br>• UCA1 knockdown downregulates KIF20A |
| An et al. [71] | • HeLa, SiHa, and ME180 cells | • Overexpression of UCA1 in all cell lines, especially ME180 cells<br>• UCA1 increases proliferation, migration, and invasion in HeLa and SiHa cells<br>• UCA1 downregulates SMARCD3 through ubiquitin-mediated proteolysis, leading to cancer progression in HeLa and ME180 cells |
| | • BALB/c-nu mice | • UCA1 promotes tumor growth by suppressing SMARCD3 |
| | • 18 CC tissues and 18 nontumor tissues<br>• CC plasma exosomes | • Overexpression of UCA1 in CC tissues and CC plasma exosomes |
| An et al. [72] | • SiHa, HeLa, CaSki, and ME180 cells | • Overexpression of UCA1 increases the proliferation and invasion of all cell lines<br>• UCA1 downregulates miR-299-3p, causing the same effects on SiHa cells |
| | • 30 CC tissues and 30 normal tissues | • Upregulation of UCA1 in CC tissues<br>• UCA1 decreases miR-299-3p and promotes tumor growth |
| Wei et al. [14] | • HeLa cells | • Overexpression of UCA1 downregulates miR-145 and increases proliferation, migration, and invasion |
| | • 109 CC tissues and normal tissues | • Upregulation of UCA1 inhibits miR-145 and accelerates cancer progression |
| Wu et al. [73] | • HEC-1B, HeLa, N3CA, HEC-1A, RL95-2, and lshikawa3H12 cells | • UCA1 overexpression in all cell lines, especially HEC-1B and HeLa<br>• UCA1 increases proliferation and glycolysis by inhibiting the miR-493-5p/HK2 axis in HEC-1B and HeLa cells |
| | • 20 CC tissues and 20 normal tissues | • Overexpression of UCA1 in 13 out of 20 CC tissues |
| Yang et al. [74] | • HeLa cells | • Overexpression of UCA1<br>• promotes EMT, proliferation, migration, and invasion by inhibiting miR-155 |
| | • 20 CC tissues and normal tissues | • UCA1 overexpression and miR-155 downregulation<br>• High UCA1 levels associated with poor survival |

## 10. Therapeutic Implications of UCA1 in CC

The following experiments have been conducted to study the participation of UCA1 in CC chemoresistance and radioresistance, respectively.

In 2017, Wang et al. were the first to identify that the established DDP-resistant HeLa cells exhibited higher UCA1 levels than their parental cells. UCA1 enhanced DDP resistance to the former cells by limiting apoptosis via caspase 3 downregulation and CDK2 upregulation, while promoting proliferation through survivin upregulation and p21 suppression [15].

Fan et al. generated the irradiation-resistant SiHa and HeLa cells (IRR), and observed that they exhibited more upregulated UCA1 expression and glycolysis compared to their parental cells, which is indicative of UCA1's capability to induce radioresistance in CC [75].

Table 8 briefly summarizes the aforementioned findings.

**Table 8.** Therapeutic implications of UCA1 in CC.

| Study | Study Model | Main Results |
|---|---|---|
| Wang et al. [15] | • HeLa<br>• DDP-resistant HeLa cells | • UCA1 overexpression confers DDP resistance<br>• UCA1 suppresses apoptosis by downregulating caspase 3 and upregulating CDK2 in DDP-resistant HeLa cells<br>• UCA1 enhances cell proliferation by increasing survivin and decreasing p21 in DDP-resistant HeLa cells |
| Fan et al. [75] | • SiHa and<br>• SiHa-IRR cells<br>• HeLa and<br>• HeLa-IRR cells | • Overexpression of UCA1 promotes radioresistance and glycolysis in both IRR cells |

## 11. Therapeutic Implications of UCA1 in VC

To date, Gao et al. are the sole research group to explore the role of UCA1 in VC, particularly in the resistance of VSCC to DDP. They found UCA1 to be upregulated in A431 and CAL-39 cells following their exposure to cancer-associated fibroblasts (CAF)-derived exosomes, and CAF-derived exosomal UCA1 mediated DDP resistance to both cell lines by sponging miR-103a and upregulating mitosis inhibitor protein kinase WEE1. In vivo, UCA1 enhanced both tumor development and DDP resistance in the mice. Finally, VSCC tissues exhibited decreased miR-103a and overexpressed UCA1 levels, with the latter being significantly correlated with advanced clinical stage and lymph node metastasis [21].

Table 9 briefly summarizes the aforementioned findings.

**Table 9.** Therapeutic implications of UCA1 in VC.

| Study | Study Model | Main Results |
|---|---|---|
| Gao et al. [21] | • A431 and CAL-39 cells | • Upregulation of UCA1 in both cells<br>• CAF-derived exosomal UCA1 increases DDP resistance to both cells through sponging miR-103a and upregulating WEE1 |
| | • 24 male BALB/c nude mice | • Exosomal UCA1 enhances tumor growth and DDP resistance |
| | • 25 VSCC tissues and normal controls | • Upregulation of UCA1 and downregulation of miR-103a in VSCC tissues<br>• Overexpression of UCA1 associated with advanced clinical stage and lymph node metastasis |

## 12. Discussion

Gynecological malignancies affect a huge amount of the female population each year. It is consequently of utmost importance to identify novel molecular markers involved in their pathogenesis, in order to develop innovative prognostic tools and therapeutic approaches. Emerging evidence has indicated that UCA1 is frequently found dysregulated in several tumors, such as bladder, colorectal, and gastric cancers [5,23]. It plays a critical role in developing previous carcinomas by modulating the proliferation, invasion, and apoptosis of cancer cells. A great number of review articles have, to date, been published on the involvement of UCA1 in the tumorigenesis of GC and its contribution to chemoresistance.

Nonetheless, no study review has been published on the concrete role of UCA1 in GC oncogenesis, prevention, and therapy. The current work, to our knowledge, constitutes the most inclusive, up-to-date review of the literature that comprehensively summarizes the numerous effects of UCA1 on GC.

The majority of the research has focused on the involvement of UCA1 in the tumorigenesis and chemoresistance of BC. The luminal-A subtype representative MCF7 and T47D cells have been included in a very large number of studies. Almost all the experiments revealed the UCA1 overexpression in both cell lines, which significantly promoted their proliferation, invasion, and migration abilities. These functions were accomplished by the sponging effect of UCA1 on several miRNAs, including miR-122-5p [24], miR-206 [33], and miR-185-5p [37], or the upregulation of UCA1 by factors such as TGF-β [29,34], HIF-1a [40], and macrophages [41]. Nevertheless, there were two studies demonstrating that MCF7 and T47D cells exhibited lower UCA1 levels than the normal BC cells [28,39]. In one of those studies, MDA-MB-231 cells, which are the most prominently featured triple-negative cells in the reviews, also manifested decreased UCA1 expression [28]. However, the rest of the reviews revealed a significant upregulation of UCA1 in the MDA-MB-231 line, leading to their enhanced invasiveness, EMT, and reduced apoptosis [22,24]. The carcinogenic impact of UCA1 on these cells was accomplished through the upregulation of other mRNAs and proteins, such as TBL1X and PTP1B, respectively, and inhibition of miRNAs, such as miRNA-143, miR-1, and miR-203 [33,34,37,38]. Notably, UCA1 was shown to be responsible for maintaining the enhanced migratory characteristics of the MDA-MB-231-derived metastatic LM2-4 cells, while the additional MDA-MB-231-derived metastatic 231BoM-1833 and 231BrM-2a variants showed a significant expression of UCA1, as well [29,42]. Several more triple-negative cell lines were included in the studies, including MCF-10AT and SUM159 cells, which exhibited considerably downregulated UCA1 expression following their exposure to the tumor suppressor Merlin. The MDA-MB-468 and HCC1937 cells also displayed decreased UCA1 levels, after their exposure to another tumor suppressor, ARID1A [26,32]. In one study, the triple-negative MDA-MB-436 and BT549 cells exhibited significantly more elevated UCA1 expression compared to the MCF7 and MDA-MB-231 cells, while another study showed the BT-20 and MDA-MB-468 cells to overexpress UCA1 after being exposed to hypoxic conditions [34,40]. The induction of apoptosis resulting from UCA1 downregulation by ARID1A was similarly observed in Her2+ JIMT1 cells, while the Her2+ SKBR3 cells demonstrated significantly increased UCA1 expression in response to hypoxic conditions, as well [32,40]. Conversely, the study reporting low UCA1 levels in MCF7, T47D, and MDA-MB-231 cells also noted downregulated UCA1 expression in SKBR3 cells [28]. Lastly, among the examined cells, only one triple-positive cell line, BT474, was studied, exhibiting upregulated UCA1 expression after its interaction with HIF-1a and macrophages, respectively [40,41].

While no experiments were conducted exclusively in vivo, the majority of the aforementioned studies utilized BC mouse xenograft models. These models served to confirm the in vitro findings by demonstrating the promotion of tumor growth and suppression of metastasis, resulting from UCA1 overexpression and its interactions with various molecular entities, such as PTP1B and SOX12, in triple-negative and Her2+ mouse xenografts [32–35].

Four research groups assessed the levels of UCA1 in the blood of BC patients. UCA1 was found to be overtly expressed in all serum and plasma samples of patients with IDC and TNBC, compared to normal and NTNBC samples, respectively. Furthermore, UCA1 concentrations were measured to be even more elevated in the blood samples of patients in advanced stages, and women with lymph node and distant organ metastatic disease [4,5]. Nonetheless, one study found that UCA1 levels were not significantly correlated with clinicopathological characteristics, whereas another study proposed that UCA1 expression was reduced in left-sided MBC tumors, suggesting a potential correlation between UCA1 levels and laterality in MBC patients [2,30]. Almost all studies identified overexpressed UCA1 levels in BC tissues compared to the healthy ones, while additionally revealing the remarkable association of high UCA1 levels with short overall survival and advanced

clinical stage disease [31,32,34,35,37,38,40,41]. One study, however, demonstrated the downregulated UCA1 expression to be significantly associated with poor prognosis for the luminal subtype BC patients [39]. Moreover, the studies revealed the enhanced ability of UCA1 to regulate the expression of various biomolecules, such as PTP1B and SOX12, and different miRNAs, such as miR-185-5p and miR-143, to further promote its oncogenic influence on BC tissues [33,35,37,38]. Interestingly, there was one study group that identified the significant correlation between high UCA1 expression and multifocality in NST invasive BC specimens [42]. Finally, one study revealed higher UCA1 expression in patients with DCIS compared to those with IDC, while another study demonstrated that UCA1 promoted bone metastasis by upregulating ADAM17 and ULBP2 [34,36]. Both findings suggest the amplified capability of UCA1 to induce migration, invasion, and tumor development in BC.

Researchers have shown significant interest in exploring the role of UCA1 in future BC treatments and its contribution to chemotherapy resistance. One study showed that UCA1 expression can be successfully suppressed in triple-negative cell lines by the Aurora kinase inhibitor CCT137690, while another study suggested that quercetin, an important phytochemical compound with anti-tumor characteristics, is able to downregulate UCA1 in MCF7 cells and subsequently cause cell cycle arrest at the G2/M phase, reducing tumor growth and triggering apoptosis [43,45]. The induction of G2 cell cycle arrest by UCA1 inhibition was replicated when MCF7 cells were transfected with shRNA UCA1 using a non-viral vesicular nanocarrier. This indicates that UCA1 RNAi may serve as a viable approach for future BC gene therapies [44]. Many studies identified the crucial role of UCA1 in mediating BC chemoresistance. For instance, UCA1 was observed to be overexpressed in Her2+ cells, leading to a notable reduction in the cells' responsiveness to trastuzumab. However, this effect was promptly reversed upon UCA1 knockdown [46]. The inhibition of UCA1 led to increased sensitivity to PTX in MCF7 cells, as well [51]. One study indicated a possible correlation between UCA1 and ADR resistance in MCF7 cells, since the ADR-resistant cells exhibited significantly more elevated UCA1 expression compared to the ADR-sensitive ones [47]. The luminal-A-type cells also exhibited significant resistance to DOX upon UCA1 overexpression, attributed to its interactions with hnRNP or TGF-$\beta$. In contrast, UCA1 knockdown enhanced the sensitivity to DOX-induced and p27-induced apoptosis, as observed in triple-negative cell lines [52,53]. Five studies analyzed the involvement of UCA1 in tamoxifen resistance, and found that the luminal-A-type LCC2 and LCC9—tamoxifen-resistant cells—as well as BT474 cells expressed significantly higher UCA1 levels compared to the MCF7 controls, whereas the UCA1 inhibition resulted in enhanced sensitivity to tamoxifen, decreased cell viability, and deceleration of tumor growth [3,23,48–50]. In the in vivo experiments, luminal-A-type BC mouse xenograft models, which were resistant to tamoxifen and PTX, exhibited a notable increase in their sensitivity to these drugs, following UCA1 silencing [23,51,52].

Finally, analysis of BC tissue specimens revealed UCA1 overexpression in all tumor samples compared to healthy samples, contributing to accelerated cancer progression [3,23,52,53]. Two of those studies found UCA1 to be upregulated in HR+ BC tissues, while one of them demonstrated that UCA1 was notably overexpressed in stage III and IV tissues, with elevated UCA1 levels identified as a prognostic biomarker for poor survival [3,23]. The same study additionally discovered that the upregulation of $\beta$-catenin by UCA1 constitutes one of the underlying mechanisms participating in tamoxifen resistance in HR+ BC tissues [23].

The influence of UCA1 on EC was studied by two research groups. UCA1 was found to be overexpressed in the patient-derived endometrioid EC cells, enhancing proliferation and EMT through the miR-143-3p/KLF and miR-1-3-p/RXFP1 pathways, respectively [8]. In addition, both type I endometrioid Ishikawa and type II adenosquamous HTB-111 cells exhibited a remarkable suppression of their migration and invasion abilities, following the inhibition of UCA1 [6].

The reduction in tumor growth and size attributed to UCA1 knockdown was also observed in mice injected with primary endometrioid EC cells transfected with the UCA1 vector, thereby corroborating the in vitro findings [8].

The analysis of tissue specimens from both studies revealed significant upregulation of UCA1 in adenocarcinoma, including endometrioid, and non-adenocarcinoma tissues compared to normal samples. This heightened expression was correlated with accelerated cancer progression and metastasis. Additionally, one study demonstrated elevated UCA1 levels in lymph node and distant metastatic tissues, with a significant association observed between high UCA1 expression and advanced-stage disease, as well as high histological grade. Both research groups identified elevated UCA1 expression as a prognostic biomarker for poor overall survival [6,8].

Ultimately, one study group revealed a substantial upregulation of UCA1 in the invasive and sphere-forming type II endometrioid HEC-50 cells exhibiting PTX resistance compared to the parental HEC-50 cells. This suggests the significant involvement of UCA1 in mediating PTX resistance [54].

A total of five research groups showed interest in the correlation of UCA1 with the pathogenesis of OC. All studied OC cells displayed a significant overexpression of UCA1, leading to their enhanced proliferation, migration, and invasion activities [55,57–59]. In some cases, the presence of UCA1 alone was able to promote the cells' growth and viability, while in others, the interaction of UCA1 with various biomolecules was needed to achieve its carcinogenic influence. For example, the sole expression of UCA1 in the serous cystadenocarcinoma SKOV3 cells and their metastatic SKOV3.ip1 variants was proven capable of increasing their survival, whereas the sponging of miR-485-5p and the upregulation of MMP14 greatly facilitated the UCA1 impact in the mucinous cystadenocarcinoma OMC685 and endometrioid adenocarcinoma A2780 cells [55,59]. Those two cell lines in particular were shown to exhibit the highest UCA1 levels out of all the cells included in the experiment [59]. Interestingly, one study demonstrated UCA1 as a super-enhancer-associated gene, particularly sensitive to treatment with the BET inhibitor (+)-JQ1 in the high-grade serous adenocarcinoma CaOV3 and UWB1.289 cells [57].

The last study additionally featured an in vivo experiment, showcasing the suppression of cancer development following the inhibition of UCA1 in the serous cystadenocarcinoma OVCA249 mouse xenograft models [57].

All analyzed OC tissues manifested UCA1 upregulation, a finding that was notably associated with FIGO staging, histological grade, peritoneal effusion, prognosis, and lymph node metastasis in the majority of cases [56–59].

The role of UCA1 as a mediator of PTX and DDP resistance in OC was the focus of research for multiple study groups. Two studies illustrated the ability of UCA1 to induce PTX resistance in SKOV3 and high-grade serous adenocarcinoma HeyA8 cells by regulating the expression of two miRNAs and their target genes: miR-129/ABCB1 and miR-654-5p/SIK, respectively [60,62]. The enhanced capacity of UCA1 to promote PTX resistance was further confirmed by observing higher UCA1 levels in serous cystadenocarcinoma PTX-resistant KFTX and KFTXlow cells, compared to the PTX-sensitive KFlow cells [61]. Another important finding of this study was the ability of UCA1 to facilitate cell-to-cell OVV spread in the SKOV3 line overexpressing UCA1, in contrast to the high-grade serous adenocarcinoma OVCAR3 and ES-2 lines displaying a lower UCA1 expression, and being more susceptible to PTX treatment, instead [61]. The results of the last study were further validated by an in vivo experiment using KFTX and KFlow mouse xenograft models [61]. Lastly, OC tissue analysis additionally verified the previously mentioned negative correlation between UCA1 and miR-654-5p, highlighting its impact on the response to PTX chemotherapy [62].

Two studies showcased the UCA1-induced DDP resistance in SKOV3 cells, accomplished through the upregulation of SRPK1 and antiapoptotic protein Bcl2, and the respective modulation of miR-143 and its target gene FOSL2 [64,66]. The downregulation of miR-143 by UCA1 conferred DDP resistance in A2780 cells, as well [64]. Conversely, silencing the short isoform of UCA1 elevated miR-27a-5p and BIM expressions, leading to enhanced sensitization of established DDP-resistant serous cystadenocarcinoma OAW42 cells to DDP [65]. The SKOV3 mouse xenografts additionally exposed the reduced DDP sensitivity and increased tumor progression caused by UCA1 overexpression [64]. Further-

more, it became evident that all scrutinized tissues exhibiting DDP resistance expressed higher levels of UCA1 compared to both DDP-sensitive and normal EOC tissues. This upregulation was often accompanied by the different expression of various biomolecules, such as SRKP1 and FOSL2 upregulation, or miR-143 downregulation [64,66]. The latter was also depicted in the analyzed OC serum-derived exosomes [64]. Although another study evaluated UCA1 as more upregulated in the DDP-resistant tissues compared to the sensitive ones, the difference did not attain statistical significance [63]. However, the same study, in conjunction with another, revealed a crucial association between UCA1 and clinicopathological features, including poor prognosis, lymph node metastasis, advanced FIGO stage, and shorter PFS [63,65].

The regulatory purpose of UCA1 In CC was studied by several groups, the majority of which examined its effects in the adenocarcinoma representative HeLa cells. All experiments revealed a significant overexpression of UCA1, leading to enhanced EMT, proliferation, and migration of the cells through the upregulation of the β-catenin/TCF-4 axis, and inhibition of miR-145, miR-493-5p, and miR-155 [14,67,70,72–74]. All additional cervical adenocarcinoma cell lines studied, including HEC-1A, Ishikawa3H12, and HEC-1B, exhibited exceptionally high levels of UCA1 expression, particularly the latter. This overexpression enhanced their malignant behavior and restrained their apoptosis [73]. UCA1 demonstrated consistent upregulation and carcinogenic effects across all SCC cells examined in the reviews. CaSki cells, for example, displayed increased proliferation and decreased apoptosis due to the upregulation of SOX2 and downregulation of miR-204 by UCA1 [69,70]. In the SiHa, Me180, and C33a cells, which are also representative of SCC of the cervix, UCA1 was found to be overexpressed and implicated in promoting tumor development [70].

Two of the referenced studies additionally conducted in vivo experiments, both demonstrating that UCA1 silencing suppressed tumor development and enhanced apoptosis in mice [69,70].

In five of the previous studies, the analysis was also conducted on CC tissue specimens, revealing significant overexpression of UCA1 in all malignant tissues. UCA1 was once again shown accountable for driving aggressive tumor growth, through its interactions with proteins such as KIF20A, and by sponging different miRNAs, such as miR-299-3p, miR-145, and miR-155 [14,70,72–74]. Finally, two research groups identified UCA1 as a factor contributing to shortened overall survival in patients, highlighting its association with poor prognosis [70,74].

Two study groups investigated the implications of UCA1 in the treatment of CC. The first one exposed the ability of UCA1 to induce DDP resistance to HeLa cells through its correlations with CDK12 and caspase 3, and the other study revealed that the overexpression of UCA1 in HeLa and SiHa cells played a pivotal role in mediating radioresistance [15,75].

Only one research group has, to date, delved into the role of UCA1 in VC. They showed exosomal UCA1 derived from CAF to induce DDP resistance to VSCC cells by suppressing miR-103a and upregulating WEE1. The in vivo experiment validated the previous findings, while the overexpression of UCA1 in the VSCC tissues was associated with advanced clinical stage disease and lymph node metastasis [21].

Unfortunately, no original research article has, to date, been published on the role of UCA1 in VGC. The lack of reviews regarding the correlation between UCA1 and VGC underlines the urgent need for future trials, in order to improve the management of female patients suffering from VGC.

Considering all these results collectively, it appears that UCA1 exerts an oncogenic role in GC, as evidenced by the majority of studies illustrating its ability to promote cancer proliferation and metastasis, and inhibit apoptosis through interactions with multiple proteins, miRNAs, and genes. Additionally, significantly elevated UCA1 expression was associated with advanced-stage disease, lymph node or distant organ metastasis, and poor overall survival, indicating its crucial value as a prognostic biomarker. The results further indicate that the upregulation of UCA1 plays a pivotal role in inducing chemoresistance,

at least partially, in all studied GC entities, implying that targeting UCA1 therapeutically holds promise for the improvement of GC management. The findings across all investigated gynecological tumors were consistent, although some results deviated from the majority. For example, while the luminal-type MCF7 cells repeatedly exhibited elevated UCA1 levels in the majority of experiments, a single study revealed downregulated UCA1 levels in both MCF7 cells and luminal subtype tissues, which additionally served as a biomarker for poor prognosis [39]. In contrast to the findings of other works, a separate study observed downregulated UCA1 expression in all analyzed BC cell types, including luminal A, triple-negative, and Her2+ cells [28]. These discrepancies probably arise from variations in methodological approaches and sample sizes, as ranging from small-scale experiments to large clinical studies impacts the reliability and applicability of the results. Differences in cell culture conditions and data analysis techniques also contribute to the variability of outcomes. Furthermore, most experiments were conducted on a pre-clinical level, and the number of studies focusing on EC, CC, and VC is insufficient compared to the research revolving around the association between UCA1 and BC. It should also be mentioned that no studies have, to date, been conducted on the impact of UCA1 in vaginal oncogenesis, and the possible involvement of UCA1 in VGC chemoresistance, despite the encouraging results displayed from the research emphasizing the rest of the GCs. All the above findings accentuate the importance of performing larger and more meticulously planned clinical studies, to more effectively evaluate the various functions of UCA1 in individuals diagnosed with all kinds of gynecological malignancies.

## 13. Conclusions

To conclude, UCA1 seems to fulfill a crucial role in the progression of most GCs, mainly by promoting the cancer cells' proliferation and migration abilities, while inhibiting apoptosis, through interacting with multiple signaling pathways and regulating the expression levels of different proteins and genes. In addition, UCA1 functions as a mediator for inducing drug resistance, therefore limiting the therapeutic efficacy of some first-line treatment options, as well as the overall survival of female patients. All this evidence strongly suggests that UCA1 could be targeted by therapeutic agents, and possibly used as a biomarker for the early diagnosis and prognosis of GC. Regardless, future randomized-controlled clinical trials are needed to test the application of such practices in the everyday management of women dealing with BC and tumors of the genital tract.

**Author Contributions:** Literature analysis and conceptualization, E.N. and I.P.; original draft preparation and writing, E.N. and I.P.; review and supervision, K.V., C.D., N.G., A.G., P.T., N.N., K.N. and I.P. All authors have read and agreed to the published version of the manuscript.

**Funding:** This research received no external funding.

**Conflicts of Interest:** The authors declare no conflicts of interest.

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
