# Peer review of "The Role of Urothelial Cancer-Associated 1 in Gynecological Cancers"

_cimb, doi:10.3390/cimb46030174_

Round 1

Reviewer 1 Report

Comments and Suggestions for Authors

In this comprehensive review, the author concisely outlines recent findings on the relevance of LncRNA UCA1 in gynecological cancers. The review emphasizes the substantial importance of UAC1 as a versatile biomarker for diagnostic, prognostic, and therapeutic purposes. While this topic is undoubtedly intriguing, there are minor issues that require resolution.

In line 45, Triple-negative breast cancer lacks estrogen receptors and progesterone receptors, not estrogen and progesterone.

The author introduces two isoforms of UCA1. What are the differences between these two, and which one is discussed in the article?

In line 120, what “tumor suppressing microRNAs (miR) proteins” means?

Author Response

In this comprehensive review, the author concisely outlines recent findings on the relevance of LncRNA UCA1 in gynecological cancers. The review emphasizes the substantial importance of UAC1 as a versatile biomarker for diagnostic, prognostic, and therapeutic purposes. While this topic is undoubtedly intriguing, there are minor issues that require resolution.

In line 45, Triple-negative breast cancer lacks estrogen receptors and progesterone receptors, not estrogen and progesterone.

Thank you for your comment on line 45 regarding the description of triple-negative breast cancer. We have now made the necessary adjustment to clarify that triple-negative breast cancer lacks estrogen receptors and progesterone receptors, rather than estrogen and progesterone themselves.

The author introduces two isoforms of UCA1. What are the differences between these two, and which one is discussed in the article?

Thank you for your comment. We now outlined the differences between the two isoforms within the text.

In line 120, what “tumor suppressing microRNAs (miR) proteins” means?

We have now improved the sentence as follows: "UCA1 facilitates the reproduction of cancer cells through interacting with tumor-suppressing microRNAs (miR) and proteins.".

Reviewer 2 Report

Comments and Suggestions for Authors

Nousiopoulou et al thoroughly summarize the literature surrounding lncRNA UCA1 in gynecological cancers. As the information in the literature review is absolutely valuable and well presented, most of my concerns are related to formatting.

Major Points:

1. Generally the formatting is incorrect for a review. To my knowledge this is not a structured review as it does not appear to follow PRISMA guidelines. So as a standard review, the subsections of "intro, materials and methods, results, and discussion" are incorrect. I recommend removing the material and methods section (and Figure 1) entirely, as a description of your literature search is not required. The "results" section thoroughly summarizes the research in each manuscript but fails to link the studies together resulting in a paper that reads much like a book report. A discussion section, while unusual, is fine, however the authors must integrate points made in their discussion throughout the body of the text. 

#2 Information in the introduction generally has inappropriate citations. While, I recognize that the authors were trying to stick to the 75 papers mentioned in their methods, the statistics of GCs need to be properly cited from original material. Going back to the current citations and find/cite the original source of the material is paramount.

Minor point:

The use of tables do not add anything new to the results text in this review. I would prefer to see a figure linking the different findings about UCA in various GMs. It may help integrate discussion points more smoothly. 

Comments on the Quality of English Language

Quality of English is fine.

Author Response

Nousiopoulou et al thoroughly summarize the literature surrounding lncRNA UCA1 in gynecological cancers. As the information in the literature review is absolutely valuable and well presented, most of my concerns are related to formatting.

Major Points:

  1. Generally the formatting is incorrect for a review. To my knowledge this is not a structured review as it does not appear to follow PRISMA guidelines. So as a standard review, the subsections of "intro, materials and methods, results, and discussion" are incorrect. I recommend removing the material and methods section (and Figure 1) entirely, as a description of your literature search is not required. The "results" section thoroughly summarizes the research in each manuscript but fails to link the studies together resulting in a paper that reads much like a book report. A discussion section, while unusual, is fine, however the authors must integrate points made in their discussion throughout the body of the text.

Thank you for your detailed feedback on the structure of review articles. The present work is indeed a narrative and not a systematic review, which, however, needs to follow the standard structure of submitted manuscripts to MDPI Journals respecting the use of subsections such as introduction, methods, results and discussion. The

material and methods section (and Figure 1) help the readership understand the way this review was conducted. The results section then presents the results of each study, while the discussion part critically analyzes them.

#2 Information in the introduction generally has inappropriate citations. While, I recognize that the authors were trying to stick to the 75 papers mentioned in their methods, the statistics of GCs need to be properly cited from original material. Going back to the current citations and find/cite the original source of the material is paramount.

Thank you for your feedback regarding the citations in the introduction. We have reviewed and corrected the references in the article to ensure appropriate citation of statistics on GCs from original sources.

 Minor point:

The use of tables do not add anything new to the results text in this review. I would prefer to see a figure linking the different findings about UCA in various GMs. It may help integrate discussion points more smoothly.

Thank you for your feedback. The tables, however, concisely, summarize the results of the different studies and help the readership conceive the main results of each section at a glance.

Reviewer 3 Report

Comments and Suggestions for Authors

Concerning the manuscript cimb-2902323 entitled “The Role of Urothelial Cancer Associated 1 in Gynecological Cancers” by Nousiopoulou et al.

As the authors state, the impact of Long-non-coding RNAs (lncRNAs) in tumor progression and metastasis is becoming increasingly recognized as being important but still relatively little is known concerning their role in the large family of gynecological cancers. In this very well written and comprehensive this review unifies the data known on the expression and role of the lncRNA urothelial Cancer Associated 1 (UCA1) across the various cancers included in this family from a total of sixty-three relevant research articles. The methodology was well described and correct. I liked how the authors presented the results  of its role in oncogenesis, proliferation and invasion going from one type of gynecological cancer to another permitting the comparison by the reader. The discussion was complete and well organized considering the relative complexity to compare all the different types of gynecological cancers.

I am very positive about this review as it is well written and presented and as stated above has very important results that were very well discussed.

I recommend immediate publication.

Author Response

Concerning the manuscript cimb-2902323 entitled “The Role of Urothelial Cancer Associated 1 in Gynecological Cancers” by Nousiopoulou et al.

As the authors state, the impact of Long-non-coding RNAs (lncRNAs) in tumor progression and metastasis is becoming increasingly recognized as being important but still relatively little is known concerning their role in the large family of gynecological cancers. In this very well written and comprehensive this review unifies the data known on the expression and role of the lncRNA urothelial Cancer Associated 1 (UCA1) across the various cancers included in this family from a total of sixty-three relevant research articles. The methodology was well described and correct. I liked how the authors presented the results of its role in oncogenesis, proliferation and invasion going from one type of gynecological cancer to another permitting the comparison by the reader. The discussion was complete and well organized considering the relative complexity to compare all the different types of gynecological cancers.

I am very positive about this review as it is well written and presented and as stated above has very important results that were very well discussed.

I recommend immediate publication.

Thank you very much for your positive feedback on our review article. We sincerely appreciate your thorough assessment of the manuscript and your recognition of the effort put into presenting the data on lncRNA UCA1 across various gynecological cancers comprehensively.

Round 2

Reviewer 2 Report

Comments and Suggestions for Authors

Thank you for revising your citations. 

The format, however, is still incorrect. I've have pasted CIMB/MDPI's format for research manuscripts (as this review is written) and review manuscripts as this review should be written. Future reviews should always be in the below format for MDPI 

  • Research manuscripts should comprise:
    • Front matter: Title, Author list, Affiliations, Abstract, Keywords.
    • Research manuscript sections: Introduction, Materials and Methods, Results, Discussion, Conclusions (optional).
    • Back matter: Supplementary Materials, Acknowledgments, Author Contributions, Conflicts of Interest, References.
  • Review manuscripts should comprise:
    • Front matter: Title, Author list, Affiliations, Abstract, Keywords.
    • Review sections: a literature review organized logically within specific sections and subsections (optional).
    • Back matter: Acknowledgments, Author Contributions, Conflicts of Interest, References.

Author Response

Thank you for your suggestions. We have now revised the paper according to the Academic Editor´s suggestions